# Microneedles in Action: Microneedling and Microneedles-Assisted Transdermal Delivery

**DOI:** 10.3390/polym14081608

**Published:** 2022-04-15

**Authors:** Dong-Jin Lim, Hong-Jun Kim

**Affiliations:** 1Department of Otolaryngology Head & Neck Surgery, University of Alabama at Birmingham, Birmingham, AL 35294-0012, USA; daniel.djlim@gmail.com; 2Department of Korean Medical Prescription, College of Korean Medicine, Woosuk University, Jeonju 54986, Korea

**Keywords:** microneedles, transdermal delivery, drug delivery, microneedling

## Abstract

Human skin is a multilayered physiochemical barrier protecting the human body. The stratum corneum (SC) is the outermost keratinized layer of skin through which only molecules with less or equal to 500 Da (Dalton) in size can freely move through the skin. Unfortunately, the conventional use of a hypothermic needle for large therapeutic agents is susceptible to needle phobia and the risk of acquiring infectious diseases. As a new approach, a microneedle (MN) can deliver therapeutically significant molecules without apparent limitations associated with its molecular size. Microneedles can create microchannels through the skin’s SC without stimulating the proprioceptive pain nerves. With recent technological advancements in both fabrication and drug loading, MN has become a versatile platform that improves the efficacy of transdermally applied therapeutic agents (TAs) and associated treatments for various indications. This review summarizes advanced fabrication techniques for MN and addresses numerous TA coating and TA elution strategies from MN, offering a comprehensive perspective on the current microneedle technology. Lastly, we discuss how microneedling and microneedle technologies can improve the clinical efficacy of a variety of skin diseases.

## 1. Introduction

Human skin is a multilayered physiochemical barrier that physically blocks the invasion of any substance mainly due to the compacted layer of about 10~15 µm of the corneal layer, also named as the skin’s stratum corneum (SC) [1,2]. The hydrophobic components such as ceramide, cholesterol, cholesterol esters, and fatty acid are intermixed with terminally differentiated keratinocytes in the corneal layer [3]. According to studies for skin permeability, only molecules with less or equal to 500 Da (Dalton) in size can freely move through the skin, whereas above 500 Da molecules such as cyclosporine (1202 Da), tacrolimus (822 Da), and ascomycin (782 Da), for example, has been mentioned as typical drugs challenging to penetrate through the normal skin [4]. Therefore, in the classical transdermal drug delivery, the 500 Dalton rule has become a gold standard for developing new compounds except for the lesion skin. However, if the therapeutic agents are somehow accepted, they have a relative advantage for better physiological actions without the hepatic first-pass metabolism [5]. To deliver chemically or enzymatically stable therapeutic agents, oral and parenteral routes would be options for administering these bioactive drugs in nano-size formulations without sacrificing the advantage of self-administration, portability, and pre-planned dosage [6,7].

However, recently developed biological drugs with a large molecular size, for example, necessitate more viable strategies that overcome the 500 Dalton rule. Not only is a tape-stripping or abrasive skin prepping pad that improves the skin’s penetration in experimental and clinical treatments, but several techniques have also been studied, including ultrasound, microneedles, iontophoresis, low-frequency sonophoresis, and electroporation to deliver therapeutic agents through the skin [8,9,10,11,12]. Among them, microneedles (MNs) have been extensively studied in the past decade (Figure 1). Since all microneedles are an array of very tiny needles in different shapes and heights in micro size, they avoid the contact of the proprioceptors in the skin, thereby offering a painless injection [13]. In a study, a small microneedle-implanted silicon chip (dimension 3 mm × 3 mm) with a height of 150 μm showed no pain when tested with human subjects [14]. Compared to the pain from a 26-gauge needle, the variations in the microneedles create 5% to 37% of the pain [15]. Microneedles can deliver large therapeutic agents through the skin safely and sustainably without compromising painless injectability. Significantly, the portability of the microneedle confers the user-friendly repeated use without pain and unwanted infections. Especially, a study regarding the potential risk of infections proved that the chance of the infection while applying MN is much less than that of a conventional hypodermic needle [16]. In combination with using appropriately manufactured MNs, it would be easily manageable, avoiding the infection risk and any misuse. In addition, a recent systemic review regarding reported side effects of microneedling concluded that microneedling is a safe treatment when used properly [17].

The purpose of this review is to provide an overview of the microneedle technique in action for delivering therapeutic agents and to provide examples of microneedle-mediated treatment for various therapeutic agents, offering the perspective of the potential of the use of microneedle.

## 2. Microneedle Technologies

This emerging technique in the transdermal delivery system enhances the skin’s permeability and allows the delivery of relatively large molecules (Figure 2). In terms of applied strength to create the micro-sized channels through the skin, only a few mechanical force is required, which is easily achieved through hands. Since its inception, there have been a variety of designs and formulations of microneedles in the market (Table 1) [19]. Because of the use of a master mold made from polydimethylsiloxane (PDMS), biodegradable polymeric microneedles have also been created [20]. With the advancement of the related technologies associated with microneedles, the current microneedles can be made from numerous materials such as silicon, silicon dioxide, polymers, glass, and other materials in addition to metallic materials. Both microneedle and microneedling techniques use various needles ranging from several hundred microns to several mili-meter in length. In general, microneedle for these techniques has a few microns in diameter to create a permeable pore through the skin with slight pain or no pain [21]. Hence, large macromolecules can travel through the instant micro-sized channels and reach out to the subcutaneous tissues, where they meet the bloodstream [22].

### 2.1. Fabrication Materials in Microneedles

#### 2.1.1. Silicone and Metals

MB Silicone is the first-generation material for creating microneedles. A dry etching technique was utilized for solid silicone MN structures [27]. Also, a wet etching technique allowed for creating silicone MN structures with a heigh of 300 µm, where the authors demonstrated that a wet etching with potassium hydroxide (KOH) produces the silicone MN structures in a reproducible manner, which minimize the processing cost and development [28]. Metals are another material type for MN structures. Stainless steel and titanium are common metals for MN structures. For example, a stainless steel MN with different lengths has been studied for transdermal drug delivery [24]. Li et al. used titanium to create a Ti porous MN array through a metal injection molding technique, suitable for mass production [29]. In delivering therapeutic peptide hormone, a titanium MN array named as Macroflux^®^ delivery system (ALZA Corporation, Mountain View, CA, USA) was studied, where desmopressin coated onto the MN array was delivered via SC and detected in the serum samples of hairless guinea pigs [30].

#### 2.1.2. Sugars

Sugars are another feasible material for the fabrication of MN structures. Molten-maltose sugar was utilized for creating innovated sugar MN structures. Miyano et al. made maltose sugar MN with different lengths, showing enough physical tolerance applicable to healthy human skin [31]. A maltose sugar MN can deliver nicardipine hydrochloride, a calcium channel blocker for hypertension. Different sugar was also used. Donnelly et al. created MN structures with galactose, another regular sugar [32]. Dextrin is another type of sugar for MN fabrication. In the dextrin-based MN structures, insulin was incorporated and applied to the skin to show the biological activity of insulin [33].

#### 2.1.3. Natural Biomacromolecules

The reverse PDMS MNs mold silk fibroin MN structures were successfully fabricated [34]. Silk is one of the extensively studied biomaterials, which has good biocompatibility and suitable mechanical property [35]. Also, silk fibroin is recognized as a safe biomaterial by the Food and Drug Administration (FDA), a good candidate for MN fabrication for therapeutic use [36]. In the silk fibroin-based MN, several vaccines for influenza, Clostridium difficile, and Shigella were successfully incorporated and made vaccination of mice, demonstrating silk fibroin MN would be an alternative option for developing transcutaneous immunization [37]. Similarly, hyaluronic acid and its salts are utilized for MN-based transcutaneous vaccination [38]. Successful transdermal insulin delivery was reported for Type 1 diabetes by incorporating insulin while fabricating a hyaluronic acid-based MN [39]. Many therapeutic agents such as peptides, proteins, immunoglobulins have been included in hyaluronic MNs [39,40,41,42]. Using the thread-forming natural polymers such as chondroitin sulfate, albumin, and dextrin, Ito et al. made erythropoietin (EPO) loaded thread-like microneedles. Through this composite material, the authors delivered the high molecular weight therapeutic protein having 34 kDa into both the bloodstream and the systemic circulation through the lymphatics in mice [43].

#### 2.1.4. Synthetic Macromolecules

The most common synthetic polymers have been utilized to create a master micro mold to fabricate MN structures. Common examples of synthetic polymers include polyvinylpyrrolidone, poly(methyl vinyl ether-co-maleic acid) (PMVE-MA), polyethylene glycol, polycarbonate (PC), and polyvinyl alcohol [44,45,46,47]. Likewise, SU-8 and poly-methyl-methacrylate (PMMA) were used to create polymeric microneedles with high-aspect ratios [48,49].

### 2.2. Structural Types in Microneedles

#### 2.2.1. Hollow Microneedles

Hollow microneedle (HM) is an array of micron-sized hypodermic needles that deliver many therapeutic agents [50]. Instead of using a conventional hypodermic needle that may bring unexpected trypanophobia or pathogenic infections during injection, for example, pre-loaded insulin in such HM could provide a quick and easy-to-use approach to deliver insulin for type 1 diabetes. Davis et al. showed the potential of providing a biologically appropriate amount of insulin for diabetes using hollow metal microneedles [51]. In this study, a single nickel-coated titanium-copper-titanium metal HM with a sharp, hollowed, and funnel-like geometry showed similar pharmacokinetics of insulin in hairless diabetic rats, comparable to that of 50 mU of insulin administered by a conventional hypothermic needle. Several improved geometric features of HMs have also been studied [52,53]. Instead of using in-plane HMs, the out-of-plane figures have been proposed, where the needles were perpendicularly presented in a two-dimensional array. Because it has two possible opening areas at the tip of the needle and at the sides of the barrel, the out-of-plan HMs can provide the desired drug exposure through the needles while minimizing the risk of clogging when applied HMs through the skin. A study showed an out-of-plan HM with different tip curvatures through a series of manufacturing processes involving deep-reactive ion etching, anisotropic wet etching, and conformal thin film deposition. Likewise, continual isotropic and anisotropic etching created gold and titanium-coated silicone HM with tapered tips [54]. According to this study, the isotropic etch allows them to make the tapered tips with 130 µm in the silicone HM array while the silicone base had a 160 µm outer diameter. After that, the silicone HMs were coated with titanium and gold to make them implantable. Because they were covered with these metals, the silicone HM had about 10-folds pressure tolerance against breaking compared to the skin’s resistive pressure. However, earlier HM designs need multiple fabrication steps with expensive and sophisticated instruments. Polymeric HMs are the more practical approach to creating hollow needles using relatively simple solvent casting methods. Mansoor et al. used a spin-coated clay-reinforced polyimide on a SU-8 pillar mold built upon a 300 μm thick Pyrex glass substrate to create robust polymeric microneedles when the polymer dried [53]. The authors then applied plasma etching or sanding techniques to make openings at the tip. Even though this polymer-based strategy is simple, the polymeric HMs showed sufficient mechanical strength for inserting them through the skin. In the market, there are two unique HMs. The AdminPen^®^ (nanobiosciences, LLC., Sunnyvale, CA, USA) is a syringe that delivers drugs through a hollow stainless-steel microneedle array [55]. According to the company’s specifications, the MN can have different lengths ranging from 600 to 1500 µm. Hence, it may cause slight pain when applied. A study describing the relationship between microneedle lengths (480, 700, 960, and 1450 µm) and normalized pain scores irrespective of the subjects’ perception of pain indicated that long MN lengths gradually in the pain scores [15]. When a 100% score was assigned to the pain of the 26-gage hypodermic needle, the authors reported that a three-fold increase in the MN lengths results in a seven-fold increase in the pain scores from 5% to 37%. Another commercial HM is the 3M Hollow Microstructured Transdermal System (*h*MTS) from 3M company, where the HM array comprises evenly spaced microneedles and a glass cartridge [56].

#### 2.2.2. Solid Microneedles

In this type of MN, high-molecular-weight compounds can be delivered through the instant microchannels made from the applied MN. A solid MN begins forming numerous microchannels on the skin’s area while using the array of solid MN and improving the skin’s permeability for a particular time. However, the created microchannels can disappear while the skin’s natural healing processes occur. Therefore, the solid MN has no second chance for adverse infectious diseases in healthy skin. In an exciting study, self-healing after insertion has been well documented. After insertion, the recovery of the damaged skin barrier has been studied with different lengths, thicknesses, widths, and several needles made of metal [57]. This study also confirmed that occlusion is the major contributing factor that delays the recovery of the skin barrier. Hence, it would expect that the detachment of the solid MN after use can facilitate the restoration of the skin barrier [57]. In early 1998, a study was performed to deliver calcein, a fluorescein complex through human skin, where the authors used a 20 × 20 array of solid MNs with sharp, 150 μm long needles [27]. According to the authors, these silicone-based solid MNs successfully delivered the low-molecular-weight compound into the subcutaneous layers of the skin by the created pores. Similarly, a study found that the microchannels created by microneedling allowed the success of insulin delivery and subsequent reduction of elevated blood glucose levels in diabetic rats [58]. Also, this study demonstrated that the transdermal administration of insulin with microneedling is comparable to the conventional needle injection. Another example of drug delivery is a Naltrexone (NTX)-loaded solid MN to manage opiate and alcohol dependence [59]. Compared to the standard NTX patch, this solid MN provided NTX over 72 h with a steady-state drug plasma concentration for 48 h. Based on this study, during the 48 h after patching, above 2 ng/mL of NTX correlated to an 85.6% narcotic blockade in human subjects intravenously administered with 25 mg of heroin. As previously noted, the solid MNs can freely deliver therapeutic agents through the tiny microchannel holes. However, the efficacy of this transdermal delivery depends on the physicochemical characteristics of the drug. For example, the molecular weight, partition coefficient, melting point, and permeability coefficient of the drug decide the extent to which the drug penetrates the SC, indicating that the general principles of skin permeation are also applicable in the case of microneedle-based transdermal delivery systems.

### 2.3. Functional Types in Microneedles

#### 2.3.1. Dissolving Microneedles

This type of MN has gained much attention because dissolving microneedles has the user-friendly feature to which the “poke and release” principle can be applied [60,61]. For dissolving the whole portion of the MN, however, these needles are typically made from polysaccharides or other polymers.

Dissolving MNs are usually made by pouring the polymeric solutions into molds and drying under a vacuum condition at ambient temperature. Before applying the polymeric solution, the therapeutic agents are co-mixed and dried together. When the dissolving MN is used, the therapeutic agents slowly dissolve into the skin while degrading the dissolving MN via a dehydration or swelling process. The most significant advantage of this type of MN is that it can deliver the agent in a single-use without happening occlusion that hampers the healing of the microchannels.

#### 2.3.2. Coated Microneedles

Instead of loading the drug in creating MNs, coated MN can apply desired medications on the surface of each pillar of the fabricated microneedles [62]. The coated needles from drug-containing dispersions can unload subdermally from the microneedle array [62]. Because this approach can use various established coating techniques elsewhere, numerous fabrication methods have been studied. They include dip coating, gas-jet drying, spray drying, electrohydrodynamic atomization (EHDA), and piezoelectric inkjet printing [63].

The most straightforward technique is dip-coating, where the microneedles are typically dipped into a drug-containing dispersion and then withdrawn. A drug-loaded thin film with different thicknesses can also be achieved on the microneedles’ surface by repetitive dip-coating. Many biomolecules, including proteins, viruses, and DNA, have been dip-coated onto microneedles for convenient transdermal delivery [64]. However, the dip-coating also creates a rounded microneedle during the slow drying process. The dried film creates the uneven and spherical covering on the top of the array’s tip and barrel of individual microneedles. To overcome this drawback, gas-jet drying can be considered [65]. In this method, a therapeutic agent can be deposited evenly upon the surface of the microneedles because the agents pass through the transitionary gas phase at the moment of the coating by a gas-jet applicator. The solution is typically formulated to have ideal surface tension and viscosity properties, allowing for the quick and even coating of the solution on the microneedle surface, thereby eliminating the potential for solution run-off, as seen with the dip-coating approach [65]. Likewise, the spray coating technique enables us to coat the microneedle with a thin and uniform film. In this process, the atomized droplets of the therapeutic-containing solution are rapidly applied on the surfaces of the microneedles to deposit and finally dried on the surface for creating an even film [66].

In the case of the EHDA-based strategy, the atomized droplets are also subjected to an applied electrical field to facilitate the atomized droplets’ rapid moving through a capillary nozzle. Then, the drop in motion can be deposited onto the microneedle electrically grounded below the nozzle tip [67]. The EHDA-based strategy can coat the only end of the MN’s tips by using an additional surface-insulated mask compared to other coating techniques. The liquid used in this procedure must have appropriate surface tension and is, therefore, usually a polymeric solution that contains a solvent, the polymer, and the active drug.

As the last technique mentioned in this review, piezoelectric inkjet printing can distribute a certain amount of therapeutic solution onto the tip of the array of MN [68,69]. Compared to other approaches, it needs a unique compounds solution with ultra-low viscosity to ensure a consistent microneedle coating. Either by applying vibrations of the solution or increasing the solutions’ temperature, this coating technique can coat a desired therapeutic agent on the tip’s surface of the microneedles.

## 3. Microneedles in Action

As described previously, MN can deliver relatively large therapeutic agents through the skin’s SC by mechanically creating a microchannel onto the skin. This easy-to-use medical approach also sheds light on the development of numerous skin-associated treatments. By integrating with other treatment modalities such as chemotherapy, radiotherapy, phototherapy, and immunotherapy, MNs have improved clinical efficacies of treatment against various indications.

### 3.1. Microneedling for Non-Lesion and Non-Cancerous Skins

Microneedling is a method to perform percutaneous collagen induction (PCI) therapy. The needles of microneedling, typically with at least several hundred micrometers in height, create non-ablative and non-confluent punctured wounds, thereby inducing a scarless healing process while inducing collagen synthesis as a part of the normal wound healing process [70]. The length of needles is dependent on the treatment indications and the treatment locations; 500 to 1000 µm needles are used for aging skin, whereas 1500 to 2000 µm needles would be preferable for scar management [71]. In contrast to microneedle-assisted transdermal delivery, microneedling needs topical anesthesia due to the relatively tall needles (e.g., 4% lidocaine cream) [72]. In vitro evaluation of microneedling indicated that there had been upregulated genes associated with tissue remodeling and wound healing. The wounds were recovered within five days post-treatment in a standardized in vitro full-thickness 3D model of human skin [73]. Hence, it is thought that micro-wounds lead to collagen and elastin production due to the involvement of expressed multiple cytokines. Acne scarring is one of the earliest cases using microneedling techniques and is being reported as an excellent dermatological indication for microneedling [74,75,76]. Interestingly, an advanced motorized microneedle, named fractional radiofrequency microneedle (FRM), can treat acne scars as well as acne vulgaris [77]. FRM may exert a thermal inhibitory effect on the sebaceous gland of acne vulgaris, thereby reducing the overproduction of sebum due to the inflammatory status of sebaceous glands [78]. Based on the same wound healing principle, microneedling is also utilized to manage facial scars, including postsurgical and posttraumatic scars [79]. Varicella scars were also another example of using microneedling-based scar management [80]. Although there are limitations to directly using it for all types of burns, microneedling would be a cost-effective option to treat hypertrophic scars after wound management [81]. In addition, striae distensae scar characterized by epidermal flattening and atrophy has been managed by needling therapy [82].

### 3.2. Microneedle-Assisted Transdermal Delivery for Cancerous Skin Diseases

In an exciting study, cisplatin, the first-line chemotherapeutic agent, was encapsulated within a lipidic nanoparticle and incorporated into a dissolvable MN to reduce its systemic toxicity and side effects [83]. After inserting into the mouse’s skin, this lipid-coated cisplatin-nanoparticles microneedles (LCC-NP-coated MNs) significantly reduced tumors in a xenograft HNSCC (head and neck squamous cancer cell) bearing mouse model while reducing tumor-associated cytotoxicity (Figure 3). Another example of utilizing the advantage of dissolvable MN in treating dermatological diseases is a photodynamic therapy in which 5-aminolevulinic acid (5-ALA) is embedded in a hyaluronic acid-based dissolvable MN [84]. In this study, a mouse breast cancer cell line called 4T1 was successfully eliminated by phototherapy using the 5-ALA loading dissolvable MN both in-vitro and in-vivo. Instead of using a self-dissolvable MN, a recent study created a light-activatable doxorubicin MN, where doxorubicin (DOX), another first-line chemotherapeutic agent, was embedded within the mixture of poly(vinyl alcohol)/polyvinylpyrrolidone and lanthanum hexaboride, which was served as a photosensitive material [85]. Under the near-infrared (NIR) light, the light-activatable DOX MN can release DOX because of the heat-generating properties of the responsive MN to NIR light, thereby demonstrating the combinational anti-cancer effects from thermal ablation (50 °C) and chemotherapy.

Besides chemotherapy, cancer immunotherapy is also a newly investigated therapeutical modality for cancers [86]. Since the skin is the first defensive line of the body containing numerous skin-resident immune cells, the concept of MN-induced microchannels on improving the human immune system is also viable [87]. A study used dissolvable MN with nano encapsulated antigens to activate dendritic cells (DCs), one of the antigen-presenting cells (APCs) residing within the skin [88]. The authors fabricated PLGA (Poly-D, L-lactide-co-glycolide) nanoparticle-containing ovalbumin (OVA) as a model antigen for effective vaccination. They delivered the antigen-holding DCs to cutaneous draining lymph via dissolvable MNs to expand antigen-specific T cells.

The efficacious delivery of antigens to antigen-presenting cells (APCs), particularly dendritic cells (DCs), and their subsequent activations remain a significant challenge in developing effective vaccines. This study highlights the potential of dissolving microneedle (MN) arrays laden with nano encapsulated antigen to increase vaccine immunogenicity by targeting antigen specifically to contiguous DC networks within the skin. Following in situ uptake, skin-resident DCs were able to deliver antigen-encapsulated poly-D, L-lactide-*co*-glycolide (PGLA) nanoparticles to cutaneous draining lymph nodes. They subsequently induced significant expansion of antigen-specific T cells. Through this MN-assisted immunotherapy, the authors demonstrated the clearance of tumors and viruses by activating antigen-specific cytotoxic CD8^+^ T cells. Not only MNs hold OVA antigens within the inner layers of the skin, but MNs also maintain the stability of OVA within the MNs. In another study, an OVA-expressing plasmid (pOVA) and immunostimulant-polyinosinic: polycytidylic acid (poly(I:C)) were complexed into cationic nanoparticles and incorporated into dissolvable MNs to improve the internalization of skin-resident APC [89].

As active immunotherapy, there has been a lot of attention on reducing tumor-induced T cell suppression [90]. Specific target proteins are programmed death protein-1 (PD-1) and cytotoxic T lymphocyte-associated antigen 4 (CTLA-4) [91]. By blocking these antigens via checkpoint blockades, the anti-tumoral activity of T cells can be resumed. Using anti-PD1 (aPD1), a study demonstrated the potential of dissolvable MN with aPD1 to treat melanoma cancer (Figure 4) [92]. The authors developed a pH-sensitive composed of aPD1, glucose oxidase (Gox) for converting blood glucose into acidic gluconic acid and dextran. Then a hyaluronic-based MN patch was utilized for delivering aPD1 in a sustainable manner dextran nanoparticle into a B16F10 mouse melanoma skin cancer model. Likewise, the same research group demonstrated the synergistic cancer treatment by providing aPD1 and IDO inhibitors via a dissolvable MN [93]. The authors also included IDO inhibitors to improve anti-tumoral activity in this study. According to cancer immunology, t indoleamine 2,3-dioxygenase (IDO) is one of the immunosuppressive molecules found in regulatory dendritic cells (DCs) and the enzyme that degrades tryptophan into several catabolites, thereby limiting T cell function [94].

## 4. Future Perspective on Microneedling and Microneedles

Microneedling and MN-assisted treatments for various skin problems, including skin cancers, are likely to cure or manage skin diseases, even in chronic cases. Here, major MNs (solid and hollow) and drug-eluting technologies were discussed through MNs (dissolving and coated). Then, clinical uses of microneedling and MN-assisted clinical modalities, especially for PDT and skin cancers, were reviewed. As noted, there have been a lot of growing new approaches where MN and other pharmaceutical technologies are utilized for creating a new clinical modality to treat a sophisticated disease (e.g., melanoma) Using thermo-responsive lanthanum hexaboride nanostructures pre-embedded in polycaprolactone MN, for example, the authors can precisely unload desired drugs into the inner part of the skin with near-infrared (NIR) light [95]. When stimulated by a NIR light, the thermo-responsive nanostructures increase the temperature of MN up to 50 °C. The light-to-heat-mediated transduction might enable us to create an on-demand MN-assisted modality to treat various skin-related problems. Similarly, a microneedle-mediated photodynamic therapy (PDT) showed improved penetration of aminolevulinic acid (ALA) accounted for a photosensitizer to increase actinic keratoses’ therapeutic efficacy has been recognized by the most common precancer associated with the photodamaged skin [96]. Such combinational approaches would be more actively addressed in future studies because both technologies have the potential to deliver therapeutic drugs with a low risk of metabolic drug clearance, providing targeted drug delivery, and allowing self-administration of medications.

Despite the remarkable advancement in materials for MN, several challenges have remained. Even though both microneedling and microneedle-assisted transdermal delivery have minimally invasive features, individual variations in side effects such as skin redness, irritation, or skin allergy are persistent. Also, the amount of loadable drug per MN patch is relatively insufficient if a specific indication may require a high dosage for a cure thereby multiple applications of MN patch would be mandatory. Such repetitive use of MN may cause secondary problems, including hypersensitivity skin reaction, elevated risk of infection, and damaged skin barriers. In this regard, a study demonstrating that the mean micropore closure time post-MN treatment (fifty stainless needles of 800 μm length) is about 60 h irrespective of the anatomical locations (upper arm, volar forearm, and abdomen) would give us a hint for the further advancement [97]. Developing advanced self-healing MN materials that expedite the closure of created wounds may become essential elements that help the healthcare industry launch more advanced MN-based therapeutic end products in the future.

## 5. Conclusions

Human Skin is a multilayered physiochemical barrier and allows the skin absorption of small molecules of less than 500 Da. As an innovated portable strategy, microneedles and microneedling have been studied to deliver therapeutic agents for curing skin disease and non-lesional skin problems. Microneedling leads to dermal remodeling, where instantly forming micro-wounds stimulate collagen and elastin production in the dermis. Whereas microneedles can deliver therapeutic agents directly into the inner part of the skin through small and tiny micro-channels in the stratum corneum (SC). This review addresses the types, materials, and modifications of microneedles (MNs) for improving the potential of MNs as a promising transdermal medical device and the clinical applications of microneedling and MN. Based on the functional benefits of both emerging technologies, new therapeutic approaches that have much more versatile features can come out in the future.

## Figures and Tables

**Figure 1 polymers-14-01608-f001:**
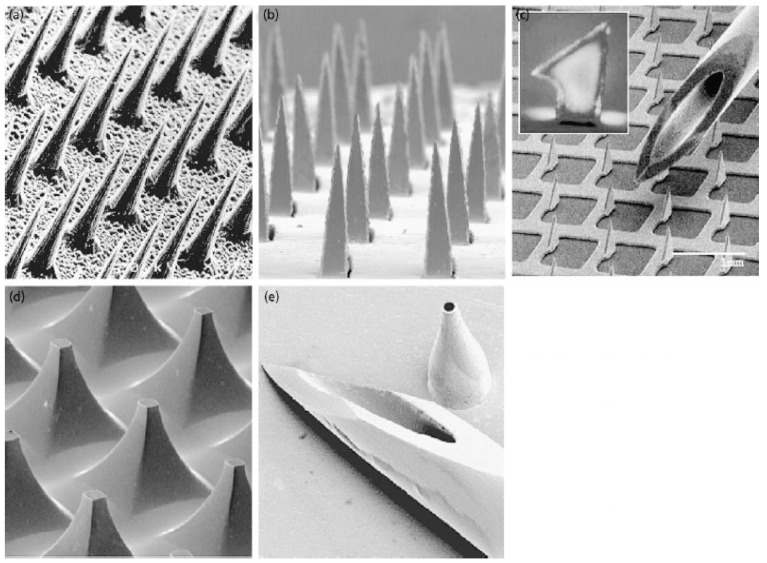
Types of microneedles used for transdermal drug delivery. (**a**) solid microneedles ionically etched from silicon wafer, (**b**) solid microneedles laser cut from stainless steel, (**c**) solid microneedles acid-etched from titanium sheet, (**d**) solid microneedles chemically etched from silicon wafers, and (**e**) hollow microneedles formed by electrodeposition of metal on to a polymer. Reproduced with permission from [18], copyright Elsevier 2004.

**Figure 2 polymers-14-01608-f002:**
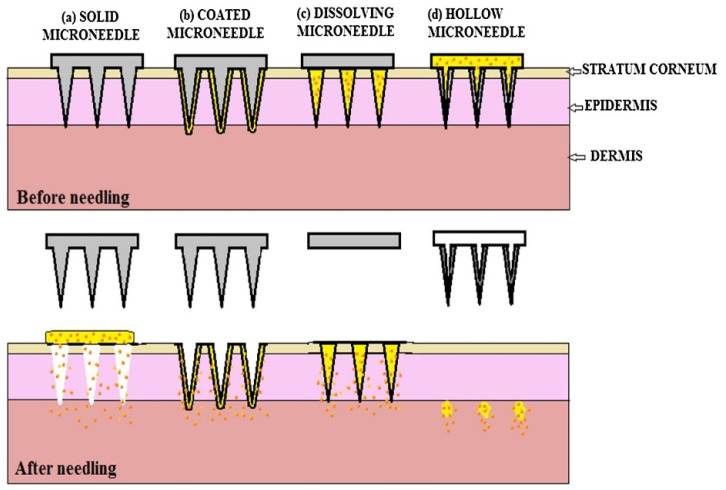
Different types of microneedles (**a**) solid microneedles use poke with patch approach, are used for pre-treatment of the skin; (**b**) coated microneedles use a coat and poke approach, a coating of drug solution is applied on the needle surface; (**c**) dissolving microneedles are made of biodegradable polymers; (**d**) hollow microneedles are filled with the drug solution and deposit the drug in the dermis. Reproduced with permission from [23], copyright Elsevier 2019.

**Figure 3 polymers-14-01608-f003:**
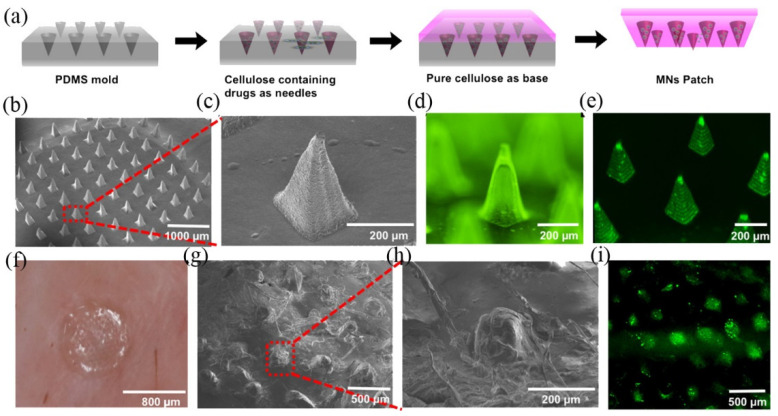
(**a**) Fabrication process of MNs. (**b**) SEM images of LCC-NP-coated MNs. (**c**) magnified SEM images of the MNs within the red box of (**b**). (**d**) fluorescence microscope image. (**e**) confocal image of NBD PE-labeled NP-coated MNs. (**f**) photograph of MN patch pressed into porcine skin. (**g**) SEM images of dissolved MN after inserting into the skin. (**h**) magnified SEM image of MNs in the red box of (**g**). (**i**) confocal image of porcine skin after treatment with NBD PE-labeled MNs. Reproduced with permission from [83]. Copyright American Chemical Society 2018.

**Figure 4 polymers-14-01608-f004:**
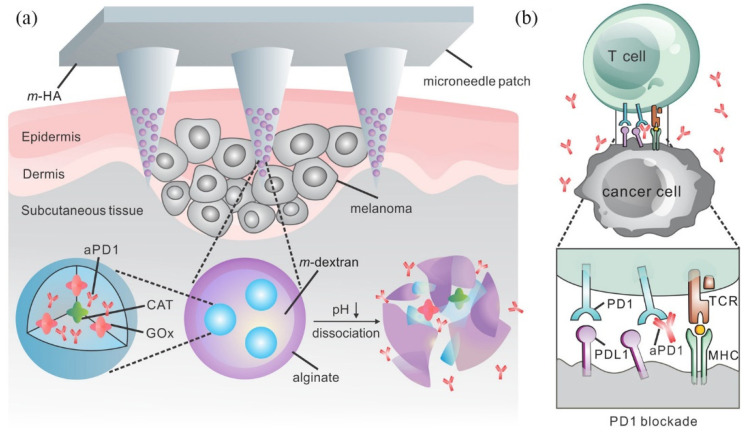
Schematic of the microneedle-assisted transdermal delivery of aPD1 for skin cancer treatment. (**a**) Schematic of the aPD1 delivered by an MN patch loaded with physiologically self-dissociated NPs. With GOx/CAT enzymatic system immobilized inside the NPs by double-emulsion method, the enzyme-mediated conversion of blood glucose to gluconic acid promotes the sustained dissociation of NPs, subsequently leading to the release of aPD1. (**b**) The blockade of PD-1 by aPD1 activates the immune system to destroy skin cancer cells. Reproduced from [92].

**Table 1 polymers-14-01608-t001:** Marketed microneedle-based transdermal products ^1^.

Brand Name	Manufacturer	Type	Application	Ref.
Dermaroller^®^	Derma Spark Product Inc., Vancouver, BC, Canada	Metallic MN Array	Used to treat acne, stretch marks, hair loss. Able to enhance drug absorption (minoxidil, hyaluronic acid, etc.).	[24]
MicroHyala^®^	CosMED Pharmaceutical Co. Ltd., Kyoto, Japan	Dissolvable MN Patch	It contains hyaluronic acid that is released into the skin to treat wrinkles.	
VaxMat^®^	TheraJect Inc., Fremont, CA, USA	Dissolvable MN Patch	It is used to deliver macromolecules, like proteins, peptides, and vaccines.	
Micro-Trans^®^	Valeritas Inc., Bridgewater, NJ, USA	Microneedle Patch	It delivers the drug into the dermis without limitations of drug size, structure, charge, or the patient’s skin characteristics.	
Drugmat^®^	TheraJect Inc., Fremont, CA, USA	Dissolvable MN Patch	It delivers hundreds of micrograms of drug rapidly through the stratum corneum into the epidermal tissue.	
Nanoject^®^	Debiotech S.A., Lausanne, Switzerland	Microneedle Array	Useful for intradermal and hypodermic drug delivery and interstitial fluid diagnostics	
Soluvia^®^	Becton Dickinson, Franklin Lakes, NJ, USA	Hollow MN Array	It is a prefillable microinjection system for accurate intradermal delivery of drugs and vaccines.	[25]
IDflu^®^/Intanza^®^	Sanofi Pasteur, Lyon, France	Intradermal MN Injection	Prefilled with influenza vaccine for intradermal influenza vaccination.	[25]
MicronJet 600^TM^	NanoPass Technologies Ltd., Nes Ziona, Israel	Intradermal MN Injection	It has been used for intradermal immunization	[25]
Micronjet^®^	NanoPass Technologies Ltd., Nes Ziona, Israel	Intradermal MN Injection	It is used with any standard syringe for painless delivery of drugs, protein, and vaccines.	
Macroflux^®^	Zosano Pharma Corp., Fremont, CA, USA	Metallic MN Array	Delivery of peptides and vaccines	[26]
Microcor^®^	Corium Inc., Boston, MA, USA	Dissolvable MN Patch	Deliver small and large molecules, like proteins, peptides, and vaccines.	
Dermapan^®^	DermapenWorld, Belrose Sydney, NSW, Australia	Microneedle Array	Used for treating various skin conditions, ranging from acne, stretch mark, and hair loss, and can enhance drug absorption.	
Microstructured Transdermal Patch	3M Company, Maplewood, MN, USA	Hollow MN Array	It delivers liquid formulations over a range of viscosities.	

^1^ This table is a revisited version of the original table shown in a reference article [19].

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
