# Peer review of "Microneedles in Action: Microneedling and Microneedles-Assisted Transdermal Delivery"

_polymers, 2022, doi:10.3390/polym14081608_

Round 1

Reviewer 1 Report

Dear Authors

The work presented here is an excellent review of a very complex and recent area of research. It has been a pleasure to read it and the relationships established here are very interesting for readers of the magazine.

I have detected in page 1, and in Line 30 where it seems to be a repetition of the word cholesterol. Among all components and compounds described, I think that there are two "cholesterols".

Congratulations for a so interesting paper

Reviewer 2 Report

Microneedles in Action: Microneedling and Microneedles-Assisted Transdermal Delivery

Dong-Jin Lim and Hong-Jun Kim

Comments to Authors:

This is an interesting review summarizing advanced fabrication techniques for microneedles, addresses coating and elution strategies, offering a comprehensive perspective on the current microneedle technology with few examples of application in clinical situation.

I have some remarks that need to be considered before publication.

General comment: for more clarity, the manuscript should only focus on application of microneedles in dermatology

  • Introduction:
  • With regard to skin permeability, examples of molecules having more than 500 Da should be considered.
  • Hypothermic needle should be corrected
  • Need to add tape stripping and dermabrasion using skin preparation pad among the several techniques studied to increase dermal penetration.
  • Regarding microneedles, it is important to give a clear explanation on different lengths used and to what application, and pain induced according to needle length
  • The example mentioned in Introduction section regarding ALA need to be moved to another section. Also, when you mention the example of ALA, you need to mention the example of MAL used in PDT.
  • Section 2
  • Need to be clear and clearly explain the relationship between needle length and the pain induced (line 86), and the need to medical professionals, particularly in the case of AdminPen with 600 – 1500 µm lengths (line 177)
  • Reversibility of holes created should be discussed with regard of potential infection

 Section 3

  • There is confusion in this section. Need to differentiate between transdermal delivery and topical application
  • Dermal remodeling: only very long needles are suitable, and this should be mentioned. Also, the use of long needles requires prior use of local anesthesia.
  • Section 4
  • This section needs to be revised, to really reflect the future perspective
  • Section 5
  • The conclusions should be carefully revised.
